# The Impact of Agricultural Insurance on Urban–Rural Income Gap: Empirical Evidence from China

**Saisai Wen** [†], **Qin Xiao** [†], **Junjie Li** and **Jianping Li** *

State Key Laboratory of Efficient Utilization of Arid and Semi-Arid Arable Land in Northern China, Institute of Agricultural Resources and Regional Planning, Chinese Academy of Agricultural Sciences, Beijing 100081, China; saisai_wen@163.com (S.W.); xiaoqin@caas.cn (Q.X.); lijunjie@caas.cn (J.L.)

* Correspondence: lijianping01@caas.cn
[†] These authors contributed equally to this work.

**Abstract:** Based on the panel data of 31 provinces in China from 2005 to 2020, this paper analyzes the mechanism and spatial spillover effect of agricultural insurance on the urban–rural income gap using a fixed effect model, an intermediary effect model, and a two-stage least square method. The results show that agricultural insurance has a significant inhibitory effect on the income gap between urban and rural areas. This inhibitory effect is realized through the path of "improving the development level of agricultural insurance-improving agricultural total factor productivity-reducing the income gap between urban and rural areas", in which the intermediary effect of agricultural total factor productivity accounts for 19.74% of the total effect. At the same time, the income gap between urban and rural areas in China exhibits typical spatial agglomeration characteristics. The western region has always been the region with the largest income gap between urban and rural areas, while the eastern region is the region with the smallest income gap between urban and rural areas. The development of agricultural insurance has had a spatial spillover effect on the income gap between urban and rural areas, and the development of agricultural insurance in neighboring areas expands the income gap between urban and rural areas in this region. In order to prevent the siphon effect, agricultural insurance investment should be increased in the neighboring regions. The results of this paper support the view of the resource flow model. Finally, we put forward some suggestions for the development of agricultural insurance, improvement in agricultural total factor productivity, and the narrowing of the income gap between urban and rural areas.

**Keywords:** agricultural insurance; TFP; urban–rural income gap; spatial agglomeration; space overflow



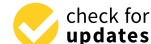

## 1. Introduction

During the era of Mao, China had the largest population in absolute poverty in the world. Given China's weak economic foundation and lack of capital, in order to swiftly enhance national strength and improve people's living standards, the Chinese government adopted a development strategy prioritizing heavy industry, drawing from the Soviet Union's development model. This strategy involved a high level of planning, with resources and investments primarily directed towards the development of the domestic heavy industry sector, thereby stimulating the growth of other production sectors. The effectiveness of this high-accumulation, high-investment, and high-energy-consumption strategy was significant, leading to rapid economic growth and industrialization within a relatively short period. Consequently, in the post-Mao reforms, China became the world champion in poverty reduction with rapid industrialization [1]. However, development does not necessarily equate to balance. This development strategy resulted in severe regional development imbalances, particularly the emergence of an urban-biased policy ideology, which continues to influence the Chinese government's governance path [2]. This urban-biased policy ideology led to various policies, including the Chinese government's

price control on agricultural products, unreasonable tax burdens, segmentation of the urban–rural labor market, and a discriminatory social welfare and security system [3], leading to a widening income gap between urban and rural areas, which has become a major challenge currently faced by China [4,5]. In order to promote rural development, the Chinese government has implemented the rural revitalization strategy, strengthening investment in rural infrastructure construction, support for rural industries, and rural cultural education. The income of Chinese farmers increased from CNY 134 in 1978 to CNY 17,131 in 2020. After deducting for inflation, rural income has increased by 18 times, growing steadily with an average annual increase of more than 7%. However, the income of urban residents in China in 1978 was CNY 343, and the income ratio between urban and rural areas was 1:2.56. In 2020, the income of urban residents in China was CNY 43,834, and the income ratio between urban and rural areas was still 1:2.56. Therefore, in the past 43 years, although the Chinese government has made great efforts to narrow the income gap between urban and rural areas, the effect is not particularly obvious. Therefore, there is still a long way to go in narrowing the gap between urban and rural areas [6,7]. According to the Statistics Bureau of China, the income gap between urban and rural residents in China has undergone an evolutionary process of narrowing–expanding–narrowing. The income ratio of urban and rural residents narrowed from 2.60:1 in 1978 to 1.82:1 in 1983 and then increased to 2.71:1 in 2017 and 2.56:1 in 2021, which is basically the same as that in 1978. Therefore, the income imbalance between urban and rural areas has become an important problem to be solved [8,9].

Agriculture is a naturally weak industry. Agricultural production is vulnerable to various external constraints and risks, and natural disasters are the key factors affecting agricultural investment. According to the Statistics Bureau of China, in 2019, various natural disasters affected 130 million people in China, and the affected area of various crops totaled 2.855 million hectares, resulting in direct agricultural economic losses of CNY 264.46 billion [10]. Drought is a meteorological disaster characterized by no rain or little rain, and it is also the most destructive natural disaster for agricultural production. Each year, China loses about 20 billion to 25 billion kg of grain due to drought, and direct economic losses amount to about CNY 15 billion to 20 billion. The flooding of rivers is caused by continuous and extremely heavy rain, and it is especially prevalent in southern China. In 2020, floods caused crop damage of 2983 thousand hectares and a loss of harvest of 516 thousand hectares [11]. As an important means of dispersing agricultural production risks, agricultural insurance is not only an important part of the national grain macro-control policy but also a green box policy advocated by the WTO to support agricultural development [8]. It plays an important role in coordinating the natural attributes of agriculture with the choice space of national macro-policy adjustment [12,13], boosting the high-quality development of agriculture and rural areas [14] and promoting the integration of China's agricultural policy design with international laws [15]. The Chinese government regards the development of agricultural insurance as an important starting point for promoting the development of agriculture and rural areas [16]. In 2022, the No.1 Document of the Central Committee of China pointed out that the insurance guarantee should be strengthened for the key rural revitalization counties in China. At the same time, the China Banking Regulatory Commission held a symposium on promoting the high-quality development of the insurance industry, emphasizing the development of inclusive insurance in relation to serving agriculture, rural areas, and farmers, strengthening the construction of financial service systems for small and micro enterprises and vulnerable groups, and implementing the policy of promoting common prosperity [17]. By 2022, China's agricultural insurance premium income reached CNY 119.2 billion, up 26.30% year-on-year, making it the country with the largest agricultural insurance premium in the world.

Agricultural insurance, as a significant means of reducing agricultural losses and stabilizing farmers' income expectations, provides a possibility to weaken the barriers of the urban–rural dual structure and narrow the income gap between urban and rural

areas. However, academic consensus has not been reached on the extent to which and how agricultural insurance can narrow the urban–rural income gap [18]. Some studies have confirmed the role of agricultural insurance in narrowing the urban–rural gap. Some scholars believe that agricultural insurance can increase farmers' income by diversifying risks, guaranteeing returns, stimulating agricultural technological progress, and encouraging large-scale farming to increase agricultural "surplus" income [19]. Others argue that agricultural insurance accelerates land transfer through land integration effects, increasing rural residents' agricultural income, property income, and non-agricultural income [20,21]. However, other research points out that the premiums of agricultural insurance, as an economic burden, often exceed the consumption ability of deeply impoverished rural households. Therefore, the poverty alleviation effect of agricultural insurance is constrained by the level of economic development [22] and can only take effect when farmers' income exceeds a certain threshold. Meanwhile, the development level of agricultural insurance varies greatly across regions, and its contribution to agricultural income also varies significantly [23]. Therefore, can agricultural insurance narrow the urban–rural gap? If it can narrow the income gap, what are the ways to do so? Agricultural insurance directly affects agricultural production and farmers' income. Some studies believe that agricultural insurance stabilizes farmers' income expectations and enhances the availability of rural credit through its risk-aversion function, which is conducive to large-scale production and specialized farming [24,25], improving the total factor productivity of agriculture. However, due to the significant information asymmetry in agricultural insurance, moral hazard and adverse selection exist, so some scholars believe that the impact of agricultural insurance on the total factor productivity of agriculture is not significant [26,27], and it may even have a negative effect [28].

Our research finds that agricultural insurance improves the total factor productivity of agriculture, thereby narrowing the urban–rural income gap. As the development stage of agricultural insurance advances, its impact on the urban–rural income gap gradually strengthens. Considering that the total factor productivity of agriculture has a high degree of spatial autocorrelation, it presents regional and agglomeration characteristics. Agricultural insurance belongs to the financial industry, and there are interactive effects in geographical space. According to the theory of financial geography, the transmission of information spillover in space has different degrees of information loss, so the information of neighboring areas is more effective, and the insurance industry in neighboring areas has a tendency to converge. Therefore, agricultural insurance is likely to have spatial spillover characteristics on the urban–rural income gap. Unlike previous studies, we incorporate agricultural insurance, the total factor productivity of agriculture, and the urban–rural income gap into a unified research framework. By further applying spatial econometric methods, we find that the agricultural insurance of surrounding areas has a siphoning effect, widening the urban–rural income gap in the local area. The reason for this is the improvement in the agricultural insurance level in surrounding areas.

The remainder of this paper is arranged as follows. The second part contains a theoretical analysis of the mechanisms whereby agricultural insurance affects agricultural total factor productivity and the urban–rural income gap; it then advances a research hypothesis. The third part describes the research design and introduces the model used in this paper, explains the selection of variables, and lists the data sources. The fourth part contains the empirical results and analysis, and the fifth part offers our conclusions and suggestions, summarizing the findings of the paper and explaining our research conclusions and policy recommendations.

## 2. Theoretical Analysis and Research Hypothesis

In the analysis of the impact of agricultural insurance on the total factor productivity (TFP) in agriculture, TFP is initially decomposed into technological progress and technical efficiency, with the latter further broken down into pure technical efficiency and scale efficiency. Hence, TFP = technological progress × technical efficiency = technological progress

$\times$ pure technical efficiency $\times$ scale efficiency. The influence of agricultural insurance on technological progress, pure technical efficiency, and scale efficiency is then sequentially analyzed through economic theory.

Agricultural insurance enhances agricultural technological progress and the scale of production by stabilizing farmers' income expectations through its risk-aversion function. Agriculture is inherently vulnerable, faced with dual risks from nature and the market. To avoid these risks and alleviate financial constraints, farmers generally do not opt for the latest agricultural technology from research departments for production, slowing down the transformation of agricultural technology achievements and the emergence of new agricultural technologies. However, agricultural insurance stabilizes farmers' income expectations, eliminating the income decline resulting from the adoption of new technologies. Therefore, farmers are more likely to adopt improved technologies to maximize profits in agricultural production [24], thereby promoting the transformation and accumulation of agricultural technology. Simultaneously, stable income expectations incentivize farmers to expand their production scale, promoting land transfers. Farmers adopt proactive production behaviors, learning and accumulating new agricultural technologies to form scale production that aligns with individual capabilities, thereby improving scale efficiency and pure technical efficiency.

Agricultural insurance accelerates agricultural technological progress and the scale of production by enhancing the availability of rural credit. Farmers' financial constraints, lack of effective collateral, and the inherent vulnerability of the agricultural industry increase farmers' credit risk, leading to long-term exclusion by formal rural financial institutions. The risk-aversion ability of agricultural insurance effectively enhances farmers' creditworthiness, thereby improving the availability of credit for farmers. The "Bank-Insurance Cooperation Model" proposed in the 2009 No.1 Central Document combines the risk-aversion ability of agricultural insurance with the fund-lending ability of rural credit, effectively alleviating farmers' financial constraints [25]. Farmers who have obtained funds tend to adopt new technologies and scale production in pursuit of maximizing profits.

Agricultural insurance accelerates agricultural technological advancement and enhances pure technical efficiency by strengthening farmers' inclination towards specialized cultivation. Traditionally, farmers have tended to diversify their crops as a way to cope with agricultural risks. Although this method stabilizes their income, it often traps small-scale farmers in a "small but complete" and "relatively closed" production environment, making it difficult to form initial accumulations for specialized and efficient agriculture. The risk-aversion function of agricultural insurance and the establishment of contracts based on the type of insurance can objectively and subjectively lead farmers towards specialized cultivation, weakening diversified planting behaviors. This is particularly evident in regions with high levels of agricultural insurance coverage. Ultimately, this leads to a dual impact of stabilizing farmers' income and improving their production efficiency. This specialized cultivation allows farmers to accumulate technology, thereby improving agricultural technological progress and technical efficiency.

Agricultural insurance enhances agricultural scale efficiency by optimizing agricultural planting structures. Governments can adjust the structure of farmers' crops through adjustments in agricultural insurance premium subsidies and the development of new types of agricultural insurance. In order to create regional agricultural product brands, extend the industrial chain, and form economies of scale, local governments often adjust the planting structure of farmers by focusing on one or a few types of crops. On one hand, agricultural insurance can influence the planting structure of farmers through income effects. By developing agricultural insurance products for specific crops (such as weather insurance and income insurance), farmers may prefer to choose crops that can be insured or crops that have high insurance benefits in order to achieve risk-aversion, thereby encouraging farmers to plant these crops on a larger scale. On the other hand, agricultural insurance can influence the planting structure of farmers through substitution effects. Governments provide different agricultural insurance subsidies for different crops,

and the relative subsidy benefits are higher than the relative income from planting competing crops. The different income levels from different crops lead farmers to optimize their planting structure, ultimately enhancing regional production scale. Agricultural insurance can promote the adjustment of crop planting areas, achieve mutual substitution between crops, and achieve the goal of optimizing the adjustment of the planting industry structure.

However, at the same time, because agricultural insurance has obvious information asymmetry characteristics, there are moral hazards and adverse selection issues, which seriously reduce agricultural technological progress and technical efficiency [26]. Therefore, some scholars believe that the impact of agricultural insurance on agricultural total factor productivity is not significant [27], and that it may even have a negative effect on agricultural total factor productivity [28]. At the same time, this paper argues that the reason for this is that the existence of moral hazard and adverse selection limits the impact of agricultural insurance on agricultural total factor productivity in the initial stages. With the improvement in agricultural insurance levels and the further standardization of insurance contracts, agricultural insurance will have a positive impact on agricultural total factor productivity.

**Hypothesis 1 (H1).** *The impact of agricultural insurance on the agricultural total factor rate is generally positive.*

**Hypothesis 2 (H2).** *The impact of agricultural total factor productivity plays an intermediary role in the impact of agricultural insurance on the income gap between urban and rural areas.*

Currently, there is limited research focusing on the spatial effects of agricultural insurance on the rural–urban income disparity. Relevant to this study is the spatial effect of inclusive insurance on poverty alleviation, where the development of inclusive insurance in a region not only reduces the incidence of poverty in that area but also in its neighboring regions [29]. Therefore, this study emphasizes theoretical derivation from a theoretical perspective.

Firstly, spatial proximity is considered. According to the first law of geography, everything is related, but near things are more related than distant things. China's agricultural population distribution has significant regional clustering characteristics. In the central and western regions, the agricultural economy occupies a high position, hence a higher proportion of the agricultural population. However, the eastern region has a higher level of urbanization, focusing on the development of industry and services, hence a lower proportion of the agricultural population. Studies have shown that in provinces with a higher proportion of agricultural population, the rural–urban income gap is larger [30]. From a spatial perspective, due to the free flow of agricultural production factors, the closer the geographical distance, the higher the flow efficiency, and therefore, neighboring areas have highly similar agricultural production. When assumption 2 (the total factor productivity of agriculture plays a mediating role in the impact of agricultural insurance on the rural–urban income gap) is proven, under similar agricultural insurance policies, the impact of agricultural insurance policies in neighboring areas on the rural–urban income gap shows spatial correlation.

Secondly, the migration of agricultural labor is considered. When the level of agricultural insurance in a region is high, farmers' income expectations are stabilized and the production scale is expanded, thereby attracting farmers from neighboring areas to move to that region. Moreover, the migrating farmers bring experience and information dissemination, improving the level of agricultural insurance in the neighboring region. Meanwhile, some studies point out that there is a siphoning phenomenon in rural finance [31], where the development of rural finance in the neighboring area can reduce rural financial investment in the local area.

Thirdly, agricultural insurance, through the "demonstration effect" and externalities, provides a basis for neighboring areas to learn and imitate, generating a spatial spillover

effect. Due to lower education and wealth levels in rural areas, the spread and utilization efficiency of agricultural insurance have been troubling developing countries. When an area increases their agricultural income due to agricultural insurance, neighboring areas will purposefully learn and adopt agricultural insurance to improve their agricultural income, which is conducive to reducing the rural–urban income gap in neighboring areas. Furthermore, the spatial spillover effect of agricultural insurance on neighboring areas is affected by the absorptive capacity of different participating entities depending on the level of human capital in rural areas.

Simultaneously, the design and promotion of agricultural insurance are often bundled with rural credit and agricultural machinery sales, requiring the joint research and development of the government, commercial banks, insurance companies, village committees, and farmers. This R&D activity is inseparable from the close connection and cooperation between the various R&D entities. This exchange accelerates the data exchange between urban and rural areas, improves the accuracy of agricultural insurance, and helps to continuously improve the quality of agricultural production and management activities. By expanding the coverage of financial services, this results in promoting China's agricultural credit funds to serve the agricultural sector, promoting the effective flow of factors in the agricultural sector, promoting the increase in income in the agricultural sector, and then promoting the overall development of agriculture to narrow the rural–urban gap [32].

**Hypothesis 3 (H3).** *Agricultural insurance has a spatial spillover effect on the urban–rural income gap, but the siphon effect or diffusion effect is uncertain.*

## 3. Research Design

### 3.1. Description of Variables

#### 3.1.1. Explained Variable

Urban–rural income gap. The ratio of the per capita disposable income of urban residents to the per capita net income of rural residents (Cirubi) is basically used to measure the urban–rural income gap in existing relevant studies [33,34]. The ratio between urban and rural per capita disposable income and rural per capita net income is used to measure the urban–rural income gap, which does not reflect the proportion of urban and rural population, while the Thiel Index considers the impact of changes in the corresponding regional population structure on the index [35]. Therefore, this paper chooses the Thiel index pair as the proxy variable of the urban–rural income gap. The greater the Thiel index, the greater the income gap between urban and rural areas. In the robustness test, the urban–rural income ratio (Cirubi) is used to measure the urban–rural income gap.

The Thiel Index is calculated as follows:

$$Theil_{it} = \sum_{j=1}^{2} \frac{y_{ijt}}{y_{it}} \ln\left( \frac{y_{ijt}}{it} \Big/ \frac{p_{ijt}}{pit} \right) \tag{1}$$

where $j$ = 1, 2 represents urban or rural areas, $y_{ijt}$ represents the per capita disposable income of the urban or rural areas in the first year of the first province, $y_{it}$ represents the total income in the first year of the first province, $t$ represents the population ($p_{ijt}$) of urban or rural areas in the first year of the first province, and $p_{it}$ represents the total population in the first year of the first province (autonomous regions and municipalities).

#### 3.1.2. Explanatory Variables

The development level of agricultural insurance is often expressed via the index of per capita premium income. Because the per capita premium income varies significantly in different regions of China, the nature and correlation of the data are not changed after taking the logarithm, but the scale of the variables is compressed. Referring to the relevant research [36], the logarithm of the per capita premium income is used to express it. In the

robustness test, the logarithm (pay) variable of per capita premium expenditure is used to measure the development level of agricultural insurance.

### 3.1.3. Mediating Variables

In this paper, the DEA–Malmquist index method is used to measure the total factor productivity of agriculture. The output index is the total output value of agriculture, forestry, animal husbandry, and fishery. In order to remove the influence of price factors, the total output value of agriculture, forestry, animal husbandry, and fishery in each region is deflated at the constant price in 2000. Regarding the input indicators, the previous research on agricultural total factor productivity commonly used six indicators: the number of members of the labor force in the primary industry, the total sown area of crops, the effective irrigation area, the amount of chemical fertilizer applied, the amount of pesticides applied, and the total power of agricultural machinery [37]. Considering that the effective irrigation area and the total sown area of crops may have multiple collinear effects, the input index no longer selects the effective irrigation area index, instead taking the number of employees in the primary industry as the labor input index, the total sown area of crops as the land input index, the total power of machinery as the machinery input index, the actual amount of agricultural chemical fertilizer as the fertilizer input index, and the effective irrigation area as the irrigation input index.

$$TFP^t\left(X^{t+1}, Y^{t+1}, X^t, Y^t\right) = \sqrt{\left[\frac{D^t(X^t, Y^t)}{D^{t+1}(X^t, Y^t)} \times \frac{D^t(X^{t+1}, Y^{t+1})}{D^{t+1}(X^{t+1}, Y^{t+1})}\right] \times \frac{D^{t+1}(X^{t+1}, Y^{t+1}/VRS)}{D^t(X^{t+1}, Y^{t+1}/VRS)} \times \frac{D^t(X^t, Y^t)}{D^{t+1}(X^{t+1}, Y^{t+1})}} \tag{2}$$

The total factor productivity obtained via the Malmquist index method is a change quantity rather than a horizontal quantity, and the calculation formula of the cumulative change rate corresponding to each index measured according to the Malmquist index is as follows:

$$ATFP_t = \prod_{t=1}^{t} TFP_t \tag{3}$$

where $TFP_t$ represents the agricultural total factor productivity index, and a $TFP_t$ greater than or less than 1 represents the growth or decline of $TFP$, respectively, that is, an increase or decrease in productivity in period $t+1$ compared with that in period $t$. $ATFP_t$ is the cumulative change value of each index in the year $TFP_t$ and the change value of each index in the $t$ year. Finally, the selected intermediary variable is the level of agricultural total factor $ATFP_t$ productivity.

### 3.1.4. Control Variables

In selecting the control variables, this paper considers the variables that have a significant influence on the income gap between urban and rural areas, including (1) the industrial structure(industry), expressed by dividing the added value of the primary industry by GDP. Generally speaking, agriculture in developing countries is predominantly labor-intensive. The larger the industrial structure value, the more rural surplus labor, the lower the labor productivity, and the greater the income gap between urban and rural areas. (2) The rural population's years of education (village edcu), rural per capita years of education = rural sample (primary school education population × 6 + junior high school population × 9 + high school population × 12 + junior college population and above × 16)/rural sample population over six years old. Under normal circumstances, people with more years of education have higher incomes. The ratio of the urban population's average years of education to the rural population's average years of education gradually narrows. (3) The agricultural disaster rate, expressed by the affected area of cultivated land/total cultivated land area. When rural areas suffer from droughts, floods, etc., agricultural income experiences a severe decline, and the income gap between urban and rural areas widens. (4) The irrigation rate, expressed by irrigated area/total cultivated land area; the improvement in farmland water conservancy levels is conducive to the

requirements of agricultural development, making it easy to form large-scale agricultural systems, and it also reflects the level of financial support for agriculture. If this metric is improved, the income gap between urban and rural areas will narrow.

Descriptive statistics of all variables are shown in Table 1.

**Table 1.** Basic results from the survey of the main variables.

| Variable | Variable Handling | Mean | Std. Dev. | Min | Max |
|---|---|---|---|---|---|
| Theil | Equation (1) | 0.106 | 0.050 | 0.018 | 0.276 |
| Cirubi | Urban disposable income/rural disposable income | 2.823 | 0.550 | 1.850 | 4.600 |
| Insurance | Ln (Agricultural insurance income/number of people employed in agriculture) | −0.628 | 2.334 | −9.009 | 3.569 |
| Pay | Ln (agricultural insurance pay/number of Agricultural labor force) | 2.236 | 2.969 | −7.278 | 9.447 |
| ATFP | Equation (2) | 1.893 | 0.853 | 0.838 | 6.421 |
| Industry | Value added of primary industry/GDP | 0.045 | 0.033 | 0.001 | 0.167 |
| Village edcu | Rural sample (primary school education population × 6 + junior high school population × 9 + high school population × 12 + junior college population and above × 16)/rural sample population over six years old | 7.460 | 0.872 | 3.238 | 9.741 |
| Disaster rate | The affected area of cultivated land/total cultivated land area | 11.36 | 13.184 | 0.000 | 80.409 |
| Irrigation rate | Irrigated land area/total cultivated land area | 0.444 | 0.2 | 0.148 | 1.234 |

### 3.2. Data Description

This paper selects the data of 31 provinces (autonomous regions and municipalities directly under the Central Government) from 2009 to 2020 for its calculations. All the data in this paper derive from the China Statistical Yearbook, China Rural Statistical Yearbook, China Financial Statistical Yearbook, China Population and Employment Statistical Yearbook, China Insurance Yearbook, China Rural Financial Services Report, provincial statistical yearbooks, the wind database, and the CSMAR database.

### 3.3. Model Building

Model Design

1.  Baseline regression. When verifying the impact of agricultural insurance on the income gap between urban and rural areas, the benchmark regression model is constructed for preliminary verification. This paper focuses on the coefficient $\alpha_1$ If the coefficient is significantly negative, this indicates that the development level of regional agricultural insurance can effectively narrow the urban–rural income gap.

$$Theil_{it} = \alpha_0 + \alpha_1 insurance + \alpha_2 Controls_{it} + \mu_i + \sigma_t + \varepsilon_{it} \tag{4}$$

where $i$ is the province; $t$ is the year; *Theil* is the explained variable, namely, the Theil index; *insurance* is the core explanatory variable, namely, the development level of agricultural insurance, which is expressed via the per capita premium income; and *Controls* is the control variable, which is expressed as the industrial structure, the ratio of the urban and rural populations for time in education, the agricultural disaster rate, and the farmland irrigation rate. $\sigma_t$ is the province fixed effect, $\varepsilon_{it}$ is the year fixed effect, and $\mu_i$ is the random error term.

2.  The mediating effect model. In order to further explore the effect of rural credit on agricultural total factor productivity in agricultural insurance, this paper draws lessons from the intermediary effect test method adopted in [37]. The mediating effect is mainly used to measure the degree of influence that the explanatory variable exerts on the explained variable indirectly through the intermediary variable. Equation (4)

takes agricultural total factor productivity as the explanatory variable and the agricultural insurance development level as the explanatory variable for the empirical test. Equation (5) takes the mediating variable rural credit level (logarithm of per capita farmers' loans) as the explanatory variable and the development level of agricultural insurance as the explanatory variable for analysis. The formula combines agricultural insurance and rural credit in the model for regression. This paper builds the following mediating effect model:

$$Theil_{it} = \alpha_0 + \alpha_1 insurance + \alpha_2 Controls_{it} + \mu_i + \sigma_t + \varepsilon_{it} \tag{5}$$

$$ATFP_{it} = \alpha_3 + \alpha_4 insurance + \alpha_5 Controls_{it} + \mu_i + \sigma_t + \varepsilon_{it} \tag{6}$$

$$Theil_{it} = \alpha_6 + \alpha_7 insurance + \alpha_8 ATFP_{it} + \alpha_9 Controls_{it} + \mu_i + \sigma_t + \varepsilon_{it} \tag{7}$$

According to the test steps of the mediating effect, if the coefficients in Equations (5)–(7) are significant and the coefficients become smaller or the confidence level drops, this indicates that a mediating effect exists. In addition, the meanings of the other variables are the same as those in Equations (1)–(4).

3. The spatial econometric model. According to the previous theory, if agricultural insurance has an impact on the urban–rural income gap via rural credit and agricultural total factor productivity, and if the relevant studies show that agricultural total factor productivity [38,39] has a high degree of spatial autocorrelation, does agricultural insurance have a spatial spillover effect on urban–rural income by means of rural credit and agricultural total factor productivity? First, the spatial econometric model can be used to test the urban–rural income gap on the basis of verifying the spatial correlation, so the Moran's I index is used to test the spatial autocorrelation:

$$I = \frac{n\sum_i \sum_j W_{ij}(X_i - \bar{x})(X_j - \bar{x})}{\left(\sum_i \sum_j W_{ij}\right)\sum_i (X_i - \bar{x})^2} \tag{8}$$

where $n$ is the total number of study areas; $X_i$ and $X_j$ are the Theil index values of provinces $i$ and $j$, respectively; $\bar{x}$ is the average value of the Theil index; and $W_{ij}$, the adjacency matrix, is taken for space. If the two provinces are adjacent, the value is assigned as 1. It is worth noting that Hainan Province has no neighboring province. This paper takes Guangdong Province, which is closest to Hainan Province, as a neighboring province (the shortest distance between the two provinces is 18 km) and carries out a Z test on the Moran's I index. Moran's I is within the range $[-1, 1]$. The closer the index is to 1, the higher the positive correlation between provinces. On the contrary, the closer the index is to $-1$, the higher the negative correlation between provinces. When Moran's I is 0, there is no spatial correlation.

When the income gap between urban and rural areas is highly autocorrelated, a spatial panel econometric model is established:

$$Y_{it} = \mu + \rho \sum_{i,j=1}^{31} W_{ij} Y_{ij} + \beta x_{it} + \varepsilon_{it} \tag{9}$$

$$Y_{it} = \mu + + \beta x_{it} + \rho \tag{10}$$

$$\varepsilon_{it} = \lambda \sum_{i,j=1}^{31} W_{ij} \varepsilon_{ij} + v \tag{11}$$

$$Y_{it} = \mu + \rho \sum_{i,j=1}^{31} W_{ij} Y_{ij} + \beta x_{it} + \theta \sum_{i,j=1}^{31} W_{ij} X_{ij} + \varepsilon_{it} \tag{12}$$

where $Y_{it}$ is the Theil coefficient matrix, $X_{it}$ is the agricultural insurance development level matrix, $W_{ij}$ is the weight of the spatial geographical weighting $\beta$, $\theta$ sum is the regression

coefficient and spatial overflow coefficient of the agricultural insurance development level, sum is the spatial autocorrelation coefficient, $\varepsilon_{it}$ is the intercept term, and $v$ is the random disturbance term. Equation (9) is a spatial panel lag model (SPLM), which is mainly used to test the spatial dependence of the Theil exponent. Equations (10) and (11) are spatial error models, which are used to test error problems such as missing explanatory variables. Equation (12) is a spatial Durbin model, which combines the advantages of the former two equations.

## 4. Empirical Results and Analysis

*4.1. Baseline Regression Results*

In order to avoid time-series correlation and heteroscedasticity problems, the clustering robust standard error is used to make estimations based on the bidirectional fixed model effect, and the regression results are shown in Table 2. Column (1) shows that the Theil index decreases by 0.009 for every 1% increase in the agricultural insurance level at a confidence level of 1% without control variables. Column (2) adds four control variables into the model, namely, the industrial structure, the education duration ratio of the urban and rural populations, the agricultural disaster rate, and the farmland irrigation rate. Column (3) and Column (4) show the fixed time effect and individual effect on the basis of Column (1) and Column (2), respectively, and the results are still significant at a confidence level of 5%, which preliminarily proves that agricultural insurance has a positive impact on narrowing the income gap between urban and rural areas. Among the control variables, the number of years of education of the rural population narrows the income gap between urban and rural areas at a confidence level of 1%, as expected. With the improvement in the rural population's education level, rural income will further increase. The irrigation rate at a confidence level of 1% enlarges the income gap between urban and rural areas, which may be due to the fact that the areas with higher irrigation rates are often located in the main grain-producing areas and depend on agricultural income; the increase in agricultural income is less than that of the urban non-agricultural industry income, resulting in an increase in the income gap between urban and rural areas.

**Table 2.** Regression results for the effects of agricultural insurance on the urban–rural income gap.

| Variable | (1) Theil | (2) Theil | (3) Theil | (4) Theil |
|---|---|---|---|---|
| Insurance | −0.009 *** | −0.005 *** | −0.002 *** | −0.001 * |
| | (0.000) | (0.000) | (0.000) | (0.000) |
| Industry | - | 0.200 ** | - | 0.051 |
| | | (0.000) | | (0.066) |
| Village edcu | - | −0.025 *** | - | −0.001 *** |
| | | (0.003) | | (0.000) |
| Disaster rate | - | −0.000 ** | - | −0.000 |
| | | (0.000) | | (0.000) |
| Irrigation rate | - | 0.010 | - | 0.047 *** |
| | | (0.011) | | (0.009) |
| _ cons | 0.099 *** | 0.278 *** | 0.123 *** | 0.115 *** |
| | (0.007) | (0.024) | (0.004) | (0.021) |
| Year effect | NO | NO | YES | YES |
| Provincial effect | NO | NO | YES | YES |
| R2 | 0.575 | 0.626 | 0.905 | 0.906 |
| N | 492 | 492 | 492 | 492 |

Note: ***, **, and * denote two-tailed *t*-tests that are statistically significant at the 1%, 5%, and 10% levels, respectively.

*4.2. Mediating Effect*

In order to further investigate whether agricultural insurance narrows the gap between urban and rural areas by improving agricultural total factor productivity, the intermediary effect of agricultural total factor productivity was tested, and the results are shown in

columns (5), (6), and (7) of Table 3. Based on the regression results presented in column (6), the regression coefficient of agricultural insurance to agricultural total factor productivity is significantly positive at the 1% confidence level, which indicates that agricultural insurance has an obvious promotion effect on agricultural total factor productivity. Based on the regression results shown in column (7), after introducing agricultural total factor productivity into the original model, the coefficient of agricultural total factor productivity is significantly positive, while the regression coefficient of agricultural insurance is significantly reduced. The test results show that agricultural total factor productivity is the intermediary of the urban–rural income gap in terms of the agricultural insurance level, and agricultural insurance promotes high-quality agricultural development by improving agricultural total factor productivity, further narrowing the urban–rural income gap. In order to test the robustness of this conclusion, the Sobel test is used to assess the mediating effect again. The results showed that, at the 10% confidence level, the mediating effect was still significant, and the mediating effect accounted for 19.74% of the total effect.

**Table 3.** Results of the mediating effect test.

| Variable | (5) Theil | (6) ATFP | (7) Theil |
|---|---|---|---|
| Insurance | −0.004 *** (0.001) | 0.209 *** (0.017) | −0.004 * (0.002) |
| ATFP | - | - | −0.003 *** (0.001) |
| Industry | 0.109 ** (0.048) | 0.960 (1.072) | 0.113 ** (0.048) |
| Village edcu | −0.036 *** (0.002) | | −0.034 (0.042) |
| Disaster rate | 0.000 (0.000) | 0.006 ** (0.002) | 0.000 (0.000) |
| Irrigation rate | −0.051 *** (0.008) | | −0.356 * (0.184) |
| _ cons | 0.392 *** (0.155) | 2.326 *** (0.347) | −0.399 *** (0.016) |
| Year effect | YES | YES | YES |
| Provincial effect | YES | YES | YES |
| R2 | 0.596 | 0.326 | 0.598 |
| N | 492 | 492 | 492 |
| Sobel test Z value | | −1.714 * | |

Note: ***, **, and * denote two-tailed *t*-tests that are statistically significant at the 1%, 5%, and 10% levels, respectively.

### 4.3. Spatial Effect Test

First, the spatial characteristics of the urban–rural income gap are preliminarily assessed. This paper uses AICGIS software to take the Theil index to evaluate 2020 as examples (shown in Figure 1). The bottom left digit represents the Thiel index; the darker the color, the larger the Thiel index and the larger the urban–rural income gap. The time trend of China's urban–rural income gap is decreasing. The Theil index ranges from [0.022, 0.276] in 2005 to [0.018, 0.144] in 2020. The average value of the Theil index decreased from 0.136 in 2005 to 0.071 in 2020, and the urban–rural income gap in northeast China narrowed most obviously. The average value of the Theil index decreased from 0.097 in 2005 to 0.048 in 2020, a decrease of 50.51%. Of course, this may also have been caused by the slowdown of economic growth in northeast China and the shrinking of the urban economic level. The western, eastern, and central regions show similar results, shrinking by 49.23%, 46.62%, and 41.77%, respectively. From a spatial perspective, the western region has always been the region with the largest income gap between urban and rural areas, with the Theil index being as high as 0.195 in 2005 and 0.099 in 2020. Meanwhile, the eastern region has always been the region with the smallest income gap between urban and

rural areas, with a Theil index of 0.079 in 2005 and as low as 0.046 in 2020, even if, by 2020, the income gap between urban and rural areas in the western region was even larger than that in the eastern region fifteen years ago. From a spatial perspective, then, the income gap between urban and rural areas is likely to exhibit spatial autocorrelation.

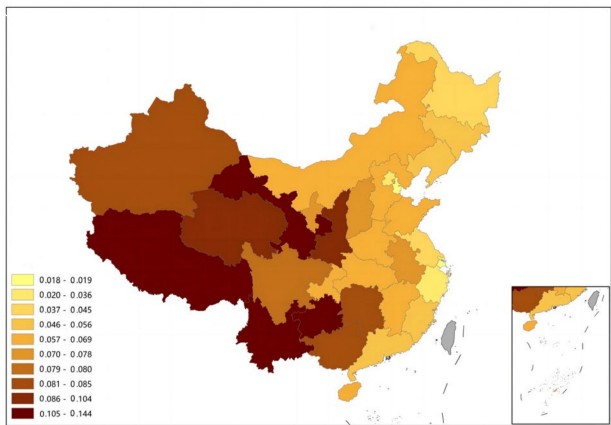

**Figure 1.** Theil index of China's provinces in and 2020.

Using Moran's I to further study the urban–rural income gap (as shown in Table 4), from 2009 to 2020, Moran's I is significantly positive, indicating that there is a significant spatial agglomeration effect in the urban–rural income gap, which has become a necessary condition for studying the spatial effect of agricultural insurance on the urban–rural income gap.

**Table 4.** Moran's I of the Theil index from 2005 to 2020.

| Variable | I | E(I) | sd(I) | z |
|----------|------|--------|-------|-----------|
| year_2005 | 0.558 | −0.033 | 0.119 | 4.993 *** |
| year_2006 | 0.559 | −0.033 | 0.119 | 4.990 *** |
| year_2007 | 0.569 | −0.033 | 0.119 | 5.070 *** |
| year_2008 | 0.580 | −0.033 | 0.119 | 5.147 *** |
| year_2009 | 0.566 | −0.033 | 0.119 | 5.036 *** |
| year_2010 | 0.559 | −0.033 | 0.119 | 4.980 *** |
| year_2011 | 0.550 | −0.033 | 0.119 | 4.912 *** |
| year_2012 | 0.544 | −0.033 | 0.118 | 4.875 *** |
| year_2013 | 0.543 | −0.033 | 0.118 | 4.868 *** |
| year_2014 | 0.548 | −0.033 | 0.118 | 4.915 *** |
| year_2015 | 0.568 | −0.033 | 0.119 | 5.070 *** |
| year_2016 | 0.563 | −0.033 | 0.119 | 5.025 *** |
| year_2017 | 0.560 | −0.033 | 0.119 | 5.008 *** |
| year_2018 | 0.547 | −0.033 | 0.118 | 4.902 *** |
| year_2019 | 0.529 | −0.033 | 0.118 | 4.763 *** |
| year_2020 | 0.510 | −0.033 | 0.118 | 4.599 *** |

Note: *** denote two-tailed *t*-tests that are statistically significant at the 1% levels, respectively.

According to the scatter charts for Moran's I in 2005 and 2020, most provinces are in the first and third quadrants, that is, they exhibit "high–high" and "low–low" clustering characteristics. Most of the western regions are in the first quadrant, and the Theil index is at its highest here, showing the agglomeration effect of the large income gap between urban and rural areas. The eastern regions and northeastern regions are concentrated in the third quadrant, where the Theil index is at its lowest, showing the agglomeration effect of the small income gap between urban and rural areas.

There is high spatial autocorrelation in the income gap between urban and rural areas. Using the method of [38] for reference, the LM test, the SDM fixed effect model, the Husman test, etc., were carried out. After the tests, it was found that the SDM model

can be simplified to SAR, and the SAR model is the best choice for the development of inclusive financing. In Table 5, looking at the core explanatory variables, in column (1), the coefficient of the level of agricultural insurance development is −0.001, which passes the test at the 5% confidence level. This validates that the development of agricultural insurance has reduced the rural–urban income disparity, which is in line with our research hypothesis that agricultural insurance enhances farmers' income by stabilizing income expectations. In column (2), the coefficient of the spatial lag term of the level of agricultural insurance development is −0.002, passing the test at the 5% confidence level. This suggests that the development of agricultural insurance in neighboring areas has widened the rural–urban income gap in the local area. This could be due to the improved level of agricultural insurance in neighboring areas, further enhancing the total factor productivity in agriculture there, thereby creating a siphoning effect that suppresses agricultural investment in the local area, affecting the income of farmers in the local area. This theory often appears in industrial research [39].

**Table 5.** Spatial Durbin model of the effect of agricultural insurance on the urban–rural income gap.

| Variable | (1) Main | (2) Wx |
|---|---|---|
| Insurance | −0.001 ** (0.000) | 0.002 ** (0.001) |
| Industry | −0.215 *** (0.051) | 0.892 *** (0.095) |
| Village edcu | −0.002 (0.002) | −0.008 * (0.003) |
| Disaster rate | −0.001 ** (0.000) | 0.000 (0.000) |
| Irrigation rate | 0.025 *** (0.006) | 0.053 *** (0.013) |
| $\rho$ | 0.616 *** (0.041) | - |
| sigma2_e | 0.000 *** (0.000) | - |
| N | 372 | - |
| $R^2$ | 0.680 | - |

Note: ***, **, and * denote two-tailed *t*-tests that are statistically significant at the 1%, 5%, and 10% levels, respectively.

The $\rho$ is significantly positive and passes the 1% significance test, indicating a spatial dependence among the rural–urban income disparities across various provinces in China. A reduction in the rural–urban income disparity in a local area leads to a reduction in the rural–urban income disparity in neighboring areas.

Further analysis of the control variables reveals that the industrial structure (industry) has reduced the rural–urban income disparity at the 1% significance level. However, at the same time, the coefficient of the spatial lag term of the industrial structure (industry) is significantly negative, indicating that the rapid increase in agricultural output value in neighboring areas creates a siphoning effect. The rapid increase in agricultural output value generally comes from increased government investment and the application of new agricultural technologies, both of which increase farmers' income and attract agricultural investment from other areas. The coefficient of the spatial lag term of the rural education level (Village edu) is significantly negative, possibly due to the knowledge spillover phenomenon. Rural residents with higher education levels are more likely to participate in agricultural insurance, learn financial knowledge, and disseminate it to neighboring areas, a point often confirmed in the economics of education [40]. The disaster rate (Disaster rate) has reduced the rural–urban income disparity at the 5% significance level. Areas with a higher disaster rate have a wider coverage of agricultural insurance. When rural areas are hit by droughts, floods, etc., agricultural income will plummet, and the rural–urban income

gap will also expand. The irrigation rate (Irrigation rate) has widened the rural–urban income disparity at the 1% significance level. The improvement in the farmland water conservancy level is beneficial to agricultural development requirements, conducive to the formation of large-scale agriculture, and also reflects the financial support for agriculture.

*4.4. Robustness Test*

1. Augmented Dickey–Fuller (ADF) test. In conducting empirical research, it is essential to first perform an Augmented Dickey–Fuller (ADF) test on the variables. This step is crucial to eliminate spurious regression caused by the non-stationarity of time-series data. In Table 6, the test results indicate that both the Theil index, the level of agricultural insurance development (insurance), and the control variables do not have a unit root at the 5% significance level, suggesting that these variables have become stationary series. This ensures that the subsequent analysis is based on reliable and valid data, enhancing the robustness of our research findings.

**Table 6.** Augmented Dickey–Fuller (ADF) test.

| Variable | 1% | 5% | ADF | Test |
|---|---|---|---|---|
| Theil | −2.460 | −2.380 | −4.3265 | Stable |
| Insurance | −2.460 | −2.380 | −7.3260 | Stable |
| Cirubi | −2.460 | −2.380 | −2.9300 | Stable |
| Pay | −2.460 | −2.380 | −6.8791 | Stable |
| Industry | −2.460 | −2.380 | −1.9714 | Stable |
| Village edcu | −2.460 | −2.380 | −6.5704 | Stable |
| Disaster rate | −2.460 | −2.380 | −8.3331 | Stable |
| Theil | −2.460 | −2.380 | −3.8066 | Stable |

2. Panel Cointegration Test. The cointegration test is used to determine whether there is a cointegrating relationship between the explanatory variables and the dependent variable, essentially determining if there is a long-term relationship among these variables. We use both the Pedroni test and the Westerlund test to perform the panel data cointegration test. The results of the Pedroni test are as follows: t (Modified Phillips-Perron) = 7.0343, t (Phillips-Perron) = −4.7491, t (Augmented Dickey–Fuller) = −3.1867. The results of the Westerlund test are as follows: Variance ratio = 4.6705. All the corresponding *p*-values of these statistics are less than 0.05. Therefore, we can reject the null hypothesis of "no cointegration" at the 5% level, suggesting that a cointegrating relationship does exist. This implies that the variables under consideration are linked over the long-term, bolstering the validity of our econometric model and the ensuing analysis.

3. Substitution variable method. The per capita premium expenditure (pay) is regarded as the proxy variable of the development level of agricultural insurance, and the results are shown in column (1) of Table 7. The development level coefficient (insurance) and per capita premium expenditure (pay) coefficient of agricultural insurance are significantly negative at the 1% confidence level, which indicates that agricultural insurance is beneficial to narrowing the gap between urban and rural areas.

4. Reducing the sample interval. There may be different samples and different empirical results. The sample interval from 2005 to 2020 is reduced to the sample interval from 2007 to 2018, and the results are shown in column (3) in Table 7. The development level coefficient of agricultural insurance is significantly negative at the 1% confidence level, which indicates that agricultural insurance is beneficial to narrowing the gap between urban and rural areas.

5. Tailing treatment. There may be outliers that produce different empirical results. The samples with the highest 1% and the lowest 1% in terms of the agricultural insurance development level are deleted, and only the middle 98% samples are retained. The results are shown in column (4) of Table 7. The development level coefficient of

agricultural insurance is significantly negative at the 10% confidence level, which indicates that agricultural insurance is beneficial to narrowing the gap between urban and rural areas.

**Table 7.** Robustness test results.

| Variable | (1) Theil | (2) Theil | (3) Theil | (4) Theil | (5) Theil |
|---|---|---|---|---|---|
| Insurance | - | −0.002 *** (0.001) | −0.041 *** (0.015) | −0.001 ** (0.000) | −0.002 *** (0.001) |
| Pay | −0.002 *** (0.001) | - | - | - | - |
| Control variable | YES | YES | YES | YES | YES |
| _ cons | 3.260 *** (0.338) | −0.141 *** (0.021) | 4.67 *** (0.022) | 0.189 ** (0.021) | 0.322 (0.014) |
| Year effect | YES | YES | YES | YES | YES |
| Provincial effect | YES | YES | YES | YES | YES |
| $R^2$ | 0.717 | 0.811 | 0.790 | 0.808 | 0.795 |
| N | 496 | 496 | 372 | 468 | 465 |

Note: *** and ** denote two-tailed *t*-tests that are statistically significant at the 1% and 5% levels, respectively.

6.  Endogenous treatment. First, there may be a causal relationship between agricultural insurance and the urban–rural income gap. Agricultural insurance narrows the urban–rural income gap by improving agricultural total factor productivity. At the same time, the narrowing of the urban–rural income gap may also release the insurance market demand. Second, the possible omission of explanatory variables and measurement errors may cause endogenous problems in this study. In this paper, 2SLS regression is employed, and the lagging agricultural insurance level is taken as the instrumental variable. The F value of the first stage is 229.93, which is significant at the confidence level of 1%, indicating that there is no weak instrumental variable. The regression results of the second stage are shown in Table 7 [5]. The coefficient of the agricultural insurance development level (insurance) is still significant at the confidence level of 1%, which shows that the model results are robust.

## 5. Discussion

This paper explores the mechanism of agricultural insurance's impact on the rural–urban income gap, finding that the total factor productivity of agriculture plays a mediating role. This result aligns with existing research [7–9], affirming the positive impact of agricultural insurance on the total factor productivity of agriculture. Distinctly, this paper decomposes the total factor productivity of agriculture into technological progress and technical efficiency, with technical efficiency further decomposed into pure technical efficiency and scale efficiency. It also theoretically analyzes the impact of the agricultural insurance's risk avoidance function, stable income expectation function, and crop structure adjustment function on technical efficiency and technological progress, thus refining the theoretical framework of agricultural insurance. Additionally, this paper employs the mediation effect model to further investigate the transmission path of "agricultural insurance—total factor productivity of agriculture—rural–urban income gap." Given the high spatial autocorrelation of the total factor productivity of agriculture, exhibiting regional and agglomeration characteristics, we further explore the spatial effects of agricultural insurance. Previous research has focused on the spatial effects of agricultural insurance on poverty reduction, treating it as part of rural finance [29]. We integrate agricultural insurance, total factor productivity of agriculture, and the rural–urban income gap into a unified research framework. Using spatial econometrics, we find that the agricultural insurance of neighboring areas has a suction effect, widening the local rural–urban income gap. This is because

the improvement in agricultural insurance levels in neighboring areas further enhances the total factor productivity of agriculture, thereby suppressing agricultural investment in the area and affecting farmers' income. Simultaneously, the rapid increase in agricultural output value in neighboring areas can generate a suction effect, attracting agricultural investment from other areas. This finding corroborates that when the increase in the agricultural output value is rapid, a suction effect still occurs, aligning with some previous research [41]. The education level of farmers in neighboring areas has a diffusion effect. Due to knowledge spillover, rural residents with higher education levels are more likely to participate in agricultural insurance, learn financial knowledge, and disseminate it to neighboring areas, reducing the local rural–urban income gap, a point often confirmed in the economics of education [42].

## 6. Conclusions and Implications

Based on the panel data of 31 provinces in China from 2005 to 2020, this paper uses a fixed effect model, an intermediary effect model, and a two-stage least square method to confirm the three hypotheses put forward by conducting robustness tests. The hypotheses were as follows: (1) The impact of agricultural insurance on the agricultural total factor rate is generally positive. (2) Agricultural total factor productivity plays an intermediary role in the impact of agricultural insurance on the urban–rural income gap, that is, it verifies the path of "improving the development level of agricultural insurance—improving agricultural total factor productivity—narrowing the urban–rural income gap". (3) Agricultural insurance has a spatial spillover effect on the income gap between urban and rural areas, and the development of agricultural insurance in neighboring areas has expanded the income gap between urban and rural areas in this region. On the basis of previous studies, this study integrates agricultural insurance, total factor productivity, and the urban–rural income gap into a unified research framework, determining the intermediary role of agricultural total factor productivity in the impact of agricultural insurance on the urban–rural income gap. On the one hand, agricultural insurance can directly narrow the urban–rural income gap through government financial subsidies; on the other hand, it can indirectly narrow the urban–rural gap by improving farmers' total factor productivity. At the same time, this paper is the first to explore the spatial relationship between agricultural insurance and the urban–rural income gap from the perspective of economic geography, which enriches the existing theories of insurance, income distribution, and economic geography. With revisions addressing the aforementioned concerns, this research could provide valuable resources for scholars, policymakers, and practitioners interested in agricultural insurance, income gap dynamics, and rural development in China.

Based on the above conclusions, we put forward the following suggestions. First, from the perspective of the government, we should enable full synergy between agricultural insurance and agricultural production. The research results show that, on the one hand, agricultural insurance, as a form of financial subsidy, directly improves farmers' incomes; on the other hand, it improves the total factor productivity of agriculture and consolidates the resilience of the rural economy. In spatial terms, it is necessary to strengthen rural financial contact and interactions in the surrounding areas, strengthen information circulation and talent and technology sharing in the surrounding areas, give full play to the spatial spillover effect of agricultural insurance, promote the process of urban–rural integration, and form a spatial agglomeration of farmers' income. Second, as far as farmers are concerned, agricultural insurance costs are basically borne by the central government and local governments. Farmers' insurance costs are far lower than commercial insurance, and their expected income is higher than that of those who do not participate in agricultural insurance and instead participate in commercial insurance. Farmers can participate in agricultural insurance in order to stabilize agricultural production expectations, improve agricultural output, and obtain more agricultural income. Finally, from the perspective of insurance companies, we should establish a diversified and efficient policy-based agricultural insurance system, optimize agricultural insurance varieties, change from traditional insur-

ance products to index insurance products, weaken moral hazards and adverse selection issues, enhance farmers' tendency to choose agricultural insurance, improve agricultural insurance efficiency, and then improve agricultural total factor productivity.

**Author Contributions:** Conceptualization, S.W. and J.L. (Junjie Liand); methodology, S.W. and J.L. (Junjie Liand); software, Q.X.; validation, S.W., Q.X. and J.L. (Jianping Li); formal analysis, S.W.; investigation, Q.X.; resources, S.W; data, S.W.; writing—original draft preparation, S.W., Q.X. and J.L. (Jianping Li); writing—review and editing, J.L. (Jianping Li); visualization, Q.X.; supervision, J.L. (Jianping Li); project administration, J.L (Jianping Li); funding acquisition, J.L. (Jianping Li). All authors have read and agreed to the published version of the manuscript.

**Funding:** This research was supported by the earmarked fund for CARS (CARS-01-54).

**Institutional Review Board Statement:** Not applicable.

**Informed Consent Statement:** Not applicable.

**Data Availability Statement:** The datasets used or analyzed during the present study are available from the corresponding authors on reasonable request.

**Conflicts of Interest:** The authors declare no conflict of interest.

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
