# Peer review of "The Impact of Agricultural Insurance on Urban–Rural Income Gap: Empirical Evidence from China"

_agriculture, doi:10.3390/agriculture13101950_

Round 1

Reviewer 1 Report

1.      The article analyzes the mechanism of the impact of agricultural insurance on the income gap between urban and rural areas. It also empirically examined the relationship between the level of development of agricultural insurance and the improvement of productivity and reduction of the income gap between urban and rural areas. These issues should be considered important and interesting for both science and economic practice.

2.      The article is well structured, however, I have comments is the style of narration. It is often vague and overly convoluted in nature. In many places, the message could be improved with decent stylistic editing. 

3.      The authors demonstrated a good knowledge of the existing literature. However, a stronger grounding in economic theories is lacking. The mention of Pigou's welfare standard is too shallow a theoretical base.

4.      The title of the article is clear and adequate.

5.      The abstract is clear, it presents the object of research, the content, and the results.

6.      The introduction defines the objective of the article, the hypotheses, and the significance of the research work.

7.      The methodology seems sound and the econometric model is correctly constructed. I have no major comments in this regard. The contributions of the paper are extensively presented in the introduction.

8.      The results and interpretations are correct but relate too poorly to the results of previous studies. It would be advisable to emphasize more strongly in the conclusion to what extent these results have added to previous knowledge. 

9.      In Section 5 Conclusions and Implications, should more strongly emphasize the fact that the research hypotheses have been confirmed. This will increase the clarity of the entire work.

The narrative is often unclear and too convoluted. In many places, the message could be improved with a decent stylistic edit.

Author Response

Response to Reviewer 1 Comments

Point 1: The article analyzes the mechanism of the impact of agricultural insurance on the income gap between urban and rural areas. It also empirically examined the relationship between the level of development of agricultural insurance and the improvement of productivity and reduction of the income gap between urban and rural areas. These issues should be considered important and interesting for both science and economic practice.

Response 1: Thanks for the reviewer's affirmation. Studying the dynamic impact and evolution of agricultural insurance on the income gap between urban and rural areas is of great significance for promoting the construction of a powerful agricultural country. How to narrow the urban-rural income gap? How to increase grain production and maintain national food security? These questions are also of interest to me. My latest research finds that agricultural insurance is conducive to narrowing the urban-rural income gap by improving agricultural total factor productivity, which has strong scientific and practical significance.

Point 2:  The article is well structured, however, I have comments is the style of narration. It is often vague and overly convoluted in nature. In many places, the message could be improved with decent stylistic editing.

Response 2: Thanks for the reviewer's suggestions. On the basis of the original rearrangement of the narrative, we re-found the English polishing agency recommended by MDPI for a new round of style editing, and now the article should be more clear.

Point 3: The authors demonstrated a good knowledge of the existing literature. However, a stronger grounding in economic theories is lacking. The mention of Pigou's welfare standard is too shallow a theoretical base.

Response 3:Thanks for the reviewer's suggestions. In the third paragraph of the introduction, I have supplemented the content of Pigouvian welfare economics and explained agricultural insurance as follows:According to Pigou's welfare economics theory, the improvement of social welfare comes from the increase in the total national income and the equalization of national distribution, that is, increasing the total income and reducing the income gap. The cost of agricultural insurance is basically borne by the central government and local governments, and the cost of farmers' insurance is far less than that of commercial insurance; their expected income will be higher than that of those who do not participate in agricultural insurance and those who participate in commercial insurance. Its influence path to social welfare is as follows: the transfer of agricultural production risk; the stabilization of farmers' income expectations; the expansion of agricultural in-vestment; industrialization (expanding the production scale and adopting new technologies); the improvement of total output and productivity; improvement in farmers' incomes; reduction in the urban–rural income gap. In other words, the Chinese government has adopted fewer financial subsidies for agricultural insurance to achieve the multiplier effect of promoting agricultural industrialization and narrowing the gap between urban and rural areas, thus driving the increase in total social welfare.

Point 4: The title of the article is clear and adequate.ï¼›The abstract is clear, it presents the object of research, the content, and the results.The introduction defines the objective of the article, the hypotheses, and the significance of the research work.The methodology seems sound and the econometric model is correctly constructed. I have no major comments in this regard. The contributions of the paper are extensively presented in the introduction.

Response 4:Thanks for the reviewer's affirmation.

Point 5: The results and interpretations are correct but relate too poorly to the results of previous studies. It would be advisable to emphasize more strongly in the conclusion to what extent these results have added to previous knowledge. In Section 5 Conclusions and Implications, should more strongly emphasize the fact that the research hypotheses have been confirmed. This will increase the clarity of the entire work.

Response 5: Thanks to the reviewer's suggestion, I re-emphasized the differences between this paper and other papers in the conclusion, and more strongly emphasized the fact that the research hypothesis has been confirmed. Now the clarity of the whole work has been improved, and the conclusion has been changed as follows:

Based on the panel data of 31 provinces in China from 2005 to 2020, this paper uses a fixed effect model, an intermediary effect model, and a two-stage least square method to confirm the three hypotheses put forward by conducting robustness tests. The hypotheses were as follows: (1) the impact of agricultural insurance on the agricultural total factor rate is generally positive. (2) Agricultural total factor productivity plays an intermediary role in the impact of agricultural insurance on the urban–rural income gap, that is, it verifies the path of "improving the development level of agricultural insurance improving agricultural total factor productivity narrowing the urban–rural income gap". (3) Agricultural insurance has spatial spillover effect on the income gap between urban and rural areas, and the development of agricultural insurance in neighboring areas has expanded the income gap between urban and rural areas in this region. On the basis of previous studies, this study integrates agricultural insurance, total factor productivity, and the urban–rural income gap into a unified research framework, determining the intermediary role of agricultural total factor productivity in the impact of agricultural insurance on urban–rural income gap. On the one hand, agricultural insurance can directly narrow the urban–rural income gap through government financial subsidies; on the other hand, it can indirectly narrow the urban–rural gap by improving farmers' total factor productivity. At the same time, this paper is the first to explore the spatial relationship between agricultural insurance and the urban–rural income gap from the perspective of economic geography, which enriches the existing theories of insurance, income distribution, and economic geography. With revisions addressing the aforementioned concerns, this research could provide valuable resources for scholars, policymakers, and practitioners interested in agricultural insurance, income gap dynamics, and rural development in China.

Based on the above conclusions, we put forward the following suggestions. First, from the perspective of the government, we should enable the full synergy between agricultural insurance and agricultural production. The research results show that, on the one hand, agricultural insurance, as a form of financial subsidy, directly improves farmers' incomes; on the other hand, it improves the total factor productivity of agriculture and consolidates the resilience of the rural economy. In spatial terms, it is necessary to strengthen rural financial contact and interactions in the surrounding areas, strengthen information circulation and talent and technology sharing in the sur-rounding areas, give full play to the spatial spillover effect of agricultural insurance, promote the process of urban–rural integration, and form a spatial agglomeration of farmers' income. Second, as far as farmers are concerned, agricultural insurance costs are basically borne by the central government and local governments. Farmers' insurance costs are far lower than commercial insurance, and their expected income is higher than that of those who do not participate in agricultural insurance and instead participate in commercial insurance. Farmers can participate in agricultural insurance in order to stabilize agricultural production expectations, improve agricultural output, and obtain more agricultural income. Finally, from the perspective of insurance companies, we should establish a diversified and efficient policy-based agricultural insurance system, optimize agricultural insurance varieties, change from traditional insurance products to index insurance products, weaken moral hazards and adverse selection issues, enhance farmers' tendency to choose agricultural insurance, improve agricultural insurance efficiency, and then improve agricultural total factor productivity.

Reviewer 2 Report

Comments:

1. The research background of this paper is a failure, the authors spent a lot of space to describe the common prosperity and agriculture, only a very limited content to describe the agricultural insurance and urban-rural income gap. I suggest that the author should pay more attention to the latter. In the introduction of this article, it's essential for the authors to deeply examine how the impact of agricultural insurance on urban-rural income gap in China. This thorough exploration should provide a comprehensive assessment of these impacts in Chinese context and incorporate academic sources to ensure alignment between the research objectives and its methodological foundation. This strategy goes beyond just setting a basic research context. Moreover, the paper should clearly convey how its findings contribute to existing academic literature, enhancing our understanding of the topic in an engaging and innovative way.

2. Contributions to articles. I think the current contribution to the existing literature is very small. I find that the findings in this paper have basically been studied by relevant scholars. Meanwhile, the contributions of this paper should be compared with previous cutting-edge and authoritative literature so as to be reflected.

3. Section 2 (2. Theoretical analysis and research hypothesis) is a failure. The theoretical basis on which the authors give their hypotheses is very weak. As for the literature analysis, the authors should conduct a disputative analysis and combine relevant theories to derive the hypotheses of this paper. It is very insufficient to present just a few pieces of literature.

4. The variable description in this paper is unclear. I suggest that the authors present a table to present these variables. Authors must clearly tell the reader how the authors deal with these variables.

5. What is the theoretical basis for choosing these variables?

6. There are models used in this paper that are not clearly explained.

7. In this paper, the basis of the statistics is missing, such as unit root test, panel heterogeneity test, correlation test and so on.

8. Relevant statistics are missing in Tables 2 to 6.

9. As pertains to the findings of this article, the authors primarily provide a descriptive account of the statistical and analytical outcomes.  It would be advisable for the authors to imbue these results with broader implications in terms of agricultural insurance on urban-rural income gap.  Additionally, elucidating the underlying rationale behind these results is warranted.  The authors should delineate the parallels and distinctions between these outcomes and those derived from the existing body of literature.

The authors are implored to delineate the paper's denouement with greater brevity and exactitude. Alongside this, the author should offer a more extensive array of pertinent policy suggestions derived directly from the paper's conclusions. Additionally, it is incumbent upon the author to outline the constraints inherent to this study and put forward plausible trajectories for subsequent academic inquiries. The existing rendition leaves room for considerable enhancement.

Extensive editing of English language required

Author Response

Response to Reviewer 2 Comments

Point 1: I have carefully reviewed the manuscript titled "The Impact of Agricultural Insurance on the Urban-Rural Income Gap: Empirical Evidence from China." Studying the dynamic impact and evolution of agricultural insurance on the income gap between urban and rural areas is of great significance for promoting the construction of a powerful agricultural country. Based on the panel data of 31 provinces in China from 2005 to 2020, this paper explores the mechanism and spatial spillover effect of agricultural insurance on the urban-rural income gap by using fixed-effect models, intermediary effect models, and the two-stage least squares method. The results show that: (1) The urban-rural income gap in China shows a decreasing trend over time, with the eastern region exhibiting the smallest gap and the northeast region the largest. (2) Agricultural insurance significantly inhibits the income gap between urban and rural areas, with agricultural total factor productivity playing a regulatory role. (3) Agricultural insurance demonstrates a spatial spillover effect on the income gap, where the development of agricultural insurance in neighboring regions can also contribute to narrowing the urban-rural income gap in the focal region. To maximize the impact of agricultural insurance, strengthening rural financial links and interactions within neighboring areas, promoting information exchange, sharing of expertise, and technology among proximate regions, and fostering urban-rural integration are essential steps." I am pleased to acknowledge the significant contributions that this manuscript presents regarding the dynamic impact of agricultural insurance on the urban-rural income gap in China. The study's focus on addressing a pressing socioeconomic concern is highly relevant and pertinent, particularly given the importance of agricultural production risk management and socioeconomic balance in fostering inclusive growth. The manuscript offers an insightful research framework, utilizing a comprehensive dataset covering 31 provinces in China over a considerable time span. This robust foundation lends credibility to the study's findings and contributes to an empirical understanding of the relationship between agricultural insurance and the urban-rural income gap. The results presented are noteworthy, indicating a declining urban-rural income gap over time and emphasizing the vital role of agricultural insurance in narrowing this gap. The regulatory role of agricultural total factor productivity adds depth to the study's analysis, shedding light on the interplay of multiple factors influencing income disparities. Moreover, the exploration of spatial spillover effects and the impact of agricultural insurance development in neighboring regions on the income gap showcases the manuscript's comprehensive approach and its potential to inform policy considerations beyond the focal region. However, I recommend a major revision of the manuscript to enhance its clarity, particularly in terms of structuring the argument and the presentation of results. Further, a more detailed discussion of the implications of the findings for policymakers and stakeholders would enrich the manuscript's contribution. Additionally, addressing potential limitations and delving into the nuances of the relationships identified would strengthen the manuscript's academic rigor. In conclusion, this manuscript possesses substantial potential to contribute to the academic literature and offer practical insights to policymakers. With revisions addressing the aforementioned concerns, this research could provide a valuable resource for scholars, policymakers, and practitioners interested in agricultural insurance, income gap dynamics, and rural development in China.

Response 1:

Thanks to the reviewer's suggestion, I re-arranged the narration in constructing the argument and the result statement. As for our research results, we conducted a study on the impact on policy makers and stakeholders from three perspectives: government, farmers and insurance companies, as follows:

First, from the perspective of the government, we should enable the full synergy be-tween agricultural insurance and agricultural production. The research results show that, on the one hand, agricultural insurance, as a form of financial subsidy, directly improves farmers' incomes; on the other hand, it improves the total factor productivity of agriculture and consolidates the resilience of the rural economy. In spatial terms, it is necessary to strengthen rural financial contact and interactions in the surrounding areas, strengthen information circulation and talent and technology sharing in the surrounding areas, give full play to the spatial spillover effect of agricultural insurance, promote the process of urbanrural integration, and form a spatial agglomeration of farmers' income. Second, as far as farmers are concerned, agricultural insurance costs are basically borne by the central government and local governments. Farmers' insur-ance costs are far lower than commercial insurance, and their expected income is higher than that of those who do not participate in agricultural insurance and instead participate in commercial insurance. Farmers can participate in agricultural insurance in order to stabilize agricultural production expectations, improve agricultural output, and obtain more agricultural income. Finally, from the perspective of insurance com-panies, we should establish a diversified and efficient policy-based agricultural insur-ance system, optimize agricultural insurance varieties, change from traditional insur-ance products to index insurance products, weaken moral hazards and adverse selec-tion issues, enhance farmers' tendency to choose agricultural insurance, improve agri-cultural insurance efficiency, and then improve agricultural total factor productivity.

In order to enhance the academic rigor of this paper, the robust test is carried out by replacing variable method, reducing sample interval, shrinking tail treatment, finding instrumental variables to deal with endogenous problems, etc. When specific parameters were changed and repeated experiments were carried out, the empirical results still did not change with the change of parameter Settings, indicating that agricultural insurance was conducive to narrowing the urban-rural gap.

Reviewer 3 Report

A review of “The impact of agricultural insurance on urban-rural income gap: Empirical evidence from China.”

Overall, I found this paper quite vague, repetitive, and unclear. There are many grammar problems and typos. This means that I am not sure if I really understood what the authors wanted to argue. As far as I understood, however, the authors wanted to argue that agricultural insurance can increase rural farmers’ productivity and income. In my opinion, this point is not convincingly discussed throughout the paper. Many points they made about the benefits of agricultural insurance were assumptive without clear references to past studies. It is not clear how this paper can shed new light as we do not know much about how past studies examined the connection between agricultural total factor productivity and agricultural insurances. Unless you are an expert on the agricultural insurance policy in China, it is impossible to know what China’ agricultural insurance cover that help stabilize income loss, property damage or farmers’ expectations. Past studies tend to argue that an agricultural insurance does not have any substantial contribution to agricultural total factor productivity. The authors believe that this point is not true if such an insurance can be improved and better standardized. However, how can this improvement and standardization be incorporated into their analyses is not clear in the methodology section. Other uncertainties exist in this paper. For example, is it possible to examine urban-rural income gaps on the basis of province-level information? Did the authors consider environmental factors, including water availability, terrains, and the soil in connection to productivity? What about rural farmers’ access to high-yielding varieties? Relatively poor rural smallholder farmers tend to be dependent on middlemen and market price fluctuations. Are there any of these things observed in China? So far, the argument the authors may want to make about the insurance-income connection might be potentially plausible but not yet convincing. I recommend major revision. I also made some comments on specific places on this paper.

Introduction

Page 1, lines 38-40: Here logic is not clear. Clarify the causal relationship between social support and income.

Page 1, line 43: Sounds redundant here. Can you be more specific?

Page 1, line 44-45: Clarify what you mean by “Chinese statistics”?

Page 2, line 45: Cite the source of this quotation. Does it have to be quoted? It does not make sense.

Page 2, line 52-55: Cite studies that support this contention, showing natural disaster loss and damage. What disasters do you have in mind here?

Page 3, lines 114-115: The authors assume that most existing studies “only examine the relationship among agricultural insurance, total factor productivity and urban-rural gap.” I am not yet convinced by this assumption. Need to cite at least some past studies.

Theoretical analysis and research hypotheses

Page 3, lines 137-138: Cite here. Otherwise, this contention is not convincing.

Page 3, lines 139-140: The authors repeatedly argue that “agricultural insurance stabilizes farmers’ income expectations and eliminates the income decline….” Whose theory is this? How can income expectations be stabilized by insurance? How can insurance eliminate income decline? Does this happen in China or elsewhere?

Page 3, line 143-144: The authors again assume that agricultural insurance accelerates technology developments. Let’s cite studies that prove this point.

Methodology

The authors argue that “with the improvement of agricultural insurance level and the further standardization of insurance contracts” they can show a positive connection between agricultural insurance and agricultural total factor productivity. Explain how this improvement and standardization were factored into analysis in the methodology section. Also, explain what agricultural insurance does for farmers in the study area (e.g., Guandong Province).

Empirical results and analysis

Page 8, lines 349-350 and lines 360-261: The authors found that the improvement of rural population’s education level further increased rural income. How is this point connected to insurance? At this point, it is not yet clear how urban-rural income gap and insurance- productivity are interconnected in this paper’s statistical analysis. Explain how agricultural total factor productivity and farmer’s income are always positively correlated. This means that the authors may prove their point if they can show that higher productivity means higher income. Is this true though?

Page 9, lines 371-372: The authors argue that “agricultural insurance promotes high-quality agricultural development by improving agricultural total factor productivity….” I want to know how the authors come to this finding. Did they find that what education level or what income level farmers are covered with insurance? To what extent those insured farmers live in rural areas. What about farmers in urban areas or peri-urban areas? Does their accessibility to the urban market help in terms of technology access and insurance coverage rate?

Pages 9-10: Figure 1 is not understandable. What do those colors and numbers represent?

Conclusions and Implications

Here it is odd to see two sections (this section and the following section) that are dedicated to conclusions. Can this section be “discussion” instead?

Page 13, lines 477-487: The authors emphasize existence of the income gap between urban and rural areas. However, this study is defined by provinces. Is it possible to define one province as urban and another one as rural? It is more convincing to see some cities are urban rather than some provinces are urban.

Conclusions

Not understandable. The first paragraph has to be more carefully phrased to make it understandable.

I made some comments in the previous section.

Author Response

Response to Reviewer 3 Comments

Introduction

Point 1:  lines 38-40: Here logic is not clear. Clarify the causal relationship between social support and income.

Response 1: The concentration of social resources contributes to the increase of income in this area. Economic development mainly refers to the transformation process of economic structure, that is, the transformation process from agricultural economy to industrial economy, specifically, it is a process of industrialization. In the process of industrialization, urbanization must develop synchronously with industrialization, which is the general law of economic development. Therefore, in the process of industrialization, the Chinese government controls the price scissors difference between workers and peasants, that is, the difference between the price of industrial products higher than the value and the price of agricultural products lower than the value, and transfers the manpower and material resources from rural areas to cities, resulting in a rapid increase in the per capita income of cities. Statistics show that in 1978, China's urban-rural income ratio was 2.39, and in 2010, China's urban-rural income gap ratio was 3.23, which has become one of the countries with the largest urban-rural income gap in the world.

Point 2: line 43: Sounds redundant here. Can you be more specific?

Response 2: In order to promote rural development, the Chinese government has implemented the rural revitalization strategy, strengthening investment in rural infrastructure construction, support for rural industries, and rural cultural education. The income of Chinese farmers has increased from 134 yuan in 1978 to 17,131 yuan in 2020. After deducting inflation, the rural income increased by 18 times, with an average annual increase of more than 7%, and the rural income grew steadily. However, the income of urban residents in China in 1978 was 343 yuan, and the income ratio between urban and rural areas was 1:2.56. In 2020, the income of urban residents in China will be 43,834 yuan, and the income ratio between urban and rural areas will still be 1:2.56. Therefore, in the past 43 years, although we have made great efforts to narrow the income gap between urban and rural areas, the effect is not particularly obvious. Therefore, narrowing the gap between urban and rural areas is a long way to go.

Point 3: line 44-45: Clarify what you mean by “Chinese statistics”?

Response3: “Chinese statistics” means according to the statistics Bureau of China.

Point 4: line 45: Cite the source of this quotation. Does it have to be quoted? It does not make sense.

Response 4: Thanks for the reviewer's suggestion. This sentence comes from the report of China National Bureau of Statistics. It does not need to be specifically quoted, so I have deleted the quote. This sentence only lists an objective fact, China's urban-rural income gap from widening - narrowing - widening - narrowing trend, behind which the reasons are:

From 1986 to 1994, as the focus of reform shifted from rural to urban areas, the income of urban residents increased significantly, and the income gap between urban and rural residents tended to widen, rising to 2.86:1 in 1994, exceeding the level in the early stage of reform and opening up. After 1995, the price of agricultural products increased farmers' incomes and the gap between urban and rural areas narrowed. However, after 1997, the gap between urban and rural areas widened significantly year by year again, from 2.9:1 in 2001 to 3.11:1 in 2002 to 3.23:1 in 2003. Only in 2004, when the country adopted a variety of measures to benefit agriculture, did the gap between urban and rural areas remain and drop to 2.56:1 in 2020. If non-monetary factors in the income of urban residents, such as housing, education, health care, social security and other social benefits, are taken into account, the income gap between urban and rural residents may be even higher.

Point 5: line 52-55: Cite studies that support this contention, showing natural disaster loss and damage. What disasters do you have in mind here?

Response 5: Drought is a meteorological disaster with no rain or little rain, and it is also the most destructive natural disaster in agricultural production. Every year, the grain loss caused by drought in China is about 20 billion ~ 25 billion kg, and the direct economic loss is about 15 billion ~ 20 billion yuan. Floods are the flooding of rivers caused by continuous heavy rains and torrential rains, especially in southern China. In 2020, floods caused 2,983 thousand hectares of crops to be affected, and 516 thousand hectares were lost (Li et al., 2011). In addition, according to FAO "Impact of Disasters and Crisis on Agriculture and Food Security", pests, diseases and disasters on crops and livestock are another major pressure on the agricultural sector. From 2008 to 2018, such biological disasters caused 9% crop and livestock production reduction.

Point 6: lines 114-115: The authors assume that most existing studies “only examine the relationship among agricultural insurance, total factor productivity and urban-rural gap.” I am not yet convinced by this assumption. Need to cite at least some past studies.

Response 6: Thanks to the reviewers for their suggestions. My words are not rigorous, so I revised this paragraph again. Most existing studies focus on agricultural insurance and total factor productivity [22-23], total factor productivity and urban-rural income gap [24, 26-27], agricultural insurance and urban-rural income gap [31-32, 35], and few studies have verified the relationship among them. On the basis of previous studies, this study integrates agricultural insurance, total factor productivity and urban-rural income gap into a unified research framework, and finds the intermediary role of agricultural total factor productivity in the impact of agricultural insurance on urban-rural income gap. On the one hand, agricultural insurance can directly narrow the urban-rural gap through government financial subsidies, on the other hand, it can indirectly narrow the urban-rural gap by improving farmers' total factor productivity. At the same time, this paper explores the spatial relationship between agricultural insurance and urban-rural income gap from the perspective of economic geography for the first time, which enriches the theories of insurance, income distribution and economic geography.

Theoretical analysis and research hypotheses

Point 7:  lines 137-138: Cite here. Otherwise, this contention is not convincing.

Response 7: Agricultural insurance stabilizes farmers' income expectations and expands agricultural production and the use of new technologies through risk avoidance function[13-14,16].

Point 8: lines 139-140: The authors repeatedly argue that “agricultural insurance stabilizes farmers’ income expectations and eliminates the income decline….” Whose theory is this? How can income expectations be stabilized by insurance? How can insurance eliminate income decline? Does this happen in China or elsewhere?

Response 8: This theory is recognized by most agricultural insurance research literatures, such as Hazell P, Varangis P. Best Practices for Subsizing Agricultural Insurance [J]. Global Food Security, 2020: 23-49. It is also mentioned in this article. According to Pigou's welfare economics theory, the improvement of social welfare comes from the increase of total national income and the equalization of national distribution. That is, increasing the total income and reducing the income gap. The cost of agricultural insurance is basically borne by the central government and local governments, and the cost of farmers' insurance is far less than that of commercial insurance, and its expected income will be higher than that of those who do not participate in agricultural insurance and those who participate in commercial insurance. Its influence path is as follows: the transfer of agricultural production risk; the stabilization of farmers' income expectations; the expansion of agricultural investment; industrialization (expanding the production scale and adopting new technologies); the improvement of total output and productivity; improvement in farmers' incomes; reduction in the urban–rural income gap. This is happening both in China (Zhu, et al) and in the United States (Yu and Smith).

Point 9: line 143-144: The authors again assume that agricultural insurance accelerates technology developments. Let’s cite studies that prove this point.

Response 9: Agricultural insurance accelerates technological development [14]

Methodology

Point 10: The authors argue that “with the improvement of agricultural insurance level and the further standardization of insurance contracts” they can show a positive connection between agricultural insurance and agricultural total factor productivity. Explain how this improvement and standardization were factored into analysis in the methodology section. Also, explain what agricultural insurance does for farmers in the study area (e.g., Guandong Province).

Response 10: This is from the theoretical analysis. Because the contract is not standardized at the beginning of agricultural insurance and farmers' low cognition, the problems of moral hazard and adverse selection are particularly serious. Once farmers are insured in insurance, they think that their crops are guaranteed, and moral hazard may occur in loss prevention and impairment, and even some people will deliberately lead to moral hazard. For example, farmers may choose to cultivate on land that is difficult to cultivate, and choose seeds with poor quality but low price. After the occurrence of livestock diseases, the failure to treat them in time increases livestock mortality and expands insurance losses, which is an important reason why early agricultural insurance has no significant or even negative impact on agricultural total factor productivity [22, 29-30]. The further standardization of insurance contracts, such as the emergence of weather index insurance, indexes the damage degree of crops caused by one or several climatic conditions (such as temperature, precipitation, wind speed, etc.), and each index has corresponding crop yield and profit and loss. The insurance contract is based on this index. When the index reaches a certain level and has a certain impact on agricultural products, the insured can get corresponding standard compensation, which greatly reduces moral hazard, and the positive effect of agricultural insurance on agricultural production is more reflected [32].

Empirical results and analysis

Point 11: lines 349-350 and lines 360-261: The authors found that the improvement of rural population’s education level further increased rural income. How is this point connected to insurance? At this point, it is not yet clear how urban-rural income gap and insurance- productivity are interconnected in this paper’s statistical analysis. Explain how agricultural total factor productivity and farmer’s income are always positively correlated. This means that the authors may prove their point if they can show that higher productivity means higher income. Is this true though?

Response 11: Higher agricultural productivity will lead to higher farmers' income. This argument can be explained by the following: Theoretically speaking, higher agricultural productivity of farmers means higher unit output. For agricultural products with large demand price elasticity, such as fruits and vegetables, price changes will bring great changes in demand, and higher unit output means higher income. For rice, wheat and other necessities of life, there is less elasticity of demand, and price changes will not bring about great changes in demand. According to market rules, higher output will bring about a sharp drop in prices, resulting in a decrease in farmers' income. However, the Chinese government will adopt the minimum purchase price measures for food crops, giving farmers a reasonable production profit. When the agricultural market price is firmly controlled by the government, higher unit output means higher income. From the empirical results, Table 3. (Results of mediating effect test) also shows that under the confidence level of 5%, the Theil index decreases by 0.003 for every unit of agricultural total factor productivity increase, which shows that agricultural total factor productivity increases farmers' income.

Point 12: lines 371-372: The authors argue that “agricultural insurance promotes high-quality agricultural development by improving agricultural total factor productivity….” I want to know how the authors come to this finding. Did they find that what education level or what income level farmers are covered with insurance? To what extent those insured farmers live in rural areas. What about farmers in urban areas or peri-urban areas? Does their accessibility to the urban market help in terms of technology access and insurance coverage rate?

Response 12: An important indicator of high-quality agricultural development is the improvement of agricultural productivity (Xia et al, 2019; Xin and An, 2019), therefore, agricultural insurance promotes high-quality agricultural development by improving agricultural total factor productivity. According to the study on influencing factors of agricultural insurance (Chen et al), people with better education level are more willing to buy agricultural insurance, and people with higher income level are also more willing to buy agricultural insurance. Because most of the agricultural insurance subsidies of the Chinese government are invested in rural areas of major grain producing areas, for example, the full cost insurance subsidies are only distributed to major grain producing counties in major grain producing areas, so the insured farmers live in rural areas to a large extent. Farmers in cities and suburbs generally buy insurance for vegetables and fruits. After obtaining agricultural insurance, they can still pass on agricultural production risks to agricultural insurance companies, stabilize agricultural income, and thus expand agricultural investment. Because urban land rent is more expensive, they are more willing to adopt new technologies to increase unit output.

[1] Xia Xianli, Chen Zhe, Zhang Huili, etc. High-quality agricultural development: digital empowerment and realization path [J]. China Rural Economy, 2019 (12): 2-15.

[2] Xin Ling, An Xiaoning. Construction and measurement analysis of evaluation system for high-quality agricultural development in China [J]. Economic Horizon, 2019 (05): 109-118. DOI: 10.16528/j.cnki.22-1054/f.201905109.

[3] Chen Yan, Ling Yuanyun, Chen Zeyu et al.An empirical study on influencing factors of purchase intention of agricultural insurance [J]. Agricultural Technology and Economics, 2007 (02): 26-30.

Point 13: Figure 1 is not understandable. What do those colors and numbers represent?

Response 13: The spatial characteristics of the urban–rural income gap are preliminarily assessed. This paper uses AICGIS software to take the Theil index to evaluate 2010 and 2020 as examples (shown in Figure 1). The bottom left digit represents the Thiel index; the darker the color, the larger the Thiel index and the larger the urban–rural income gap. The time trend of China's urban–rural income gap is decreasing.

Conclusions and Implications

Point 14: Here it is odd to see two sections (this section and the following section) that are dedicated to conclusions. Can this section be “discussion” instead?

Response 14: Thanks for the reviewer's suggestion. According to your suggestion, I will change the last section to a “discussion”.

Point 15: lines 477-487: The authors emphasize existence of the income gap between urban and rural areas. However, this study is defined by provinces. Is it possible to define one province as urban and another one as rural? It is more convincing to see some cities are urban rather than some provinces are urban.

Response 15:Thanks to the reviewers' suggestions, China's provinces include both cities and rural areas. Even if developed provinces are defined as cities, it is difficult to define underdeveloped provinces as rural areas. Because if the province is defined as a city, the income of this province itself has two parts: farmers' income and urban income, that is, agricultural income and industrial income are integrated into urban income, which has a big error. At the same time, the error of agricultural total factor productivity will become larger, and the final result is unconvincing.

Discussion :

Point 16: Not understandable. The first paragraph has to be more carefully phrased to make it understandable.

Response 16:Thanks for the reviewer's suggestion. According to your suggestion, I have revised the last section again. This paper explores the mechanism of the impact of agricultural insurance on the urban–rural income gap, and empirically analyzes the spatial and mediating effects of the urban–rural income gap based on provincial panel data from China. The main in-novations of this paper are as follows. First, the paper offers a theoretical analysis of the mechanism of the impact of agricultural insurance on the income gap between ur-ban and rural areas, and empirically analyzes the feasibility of the path of "improving the development level of agricultural insurance—improving agricultural total factor productivity—reducing the income gap between urban and rural areas". It also inves-tigates the role of agricultural total factor productivity in the impact of agricultural insurance on the income gap between urban and rural areas. Second, we find that the income gap between urban and rural areas in China is highly autocorrelated. The in-come gap between urban and rural areas in eastern China is relatively small, and the income gap between urban and rural areas in western China is relatively high, exhib-iting significant heterogeneity. At the same time, this paper explores the spatial rela-tionship between agricultural insurance and the urban–rural income gap from the perspective of economic geography for the first time, enriching the existing theories of insurance, income distribution, and economic geography.

Round 2

Reviewer 2 Report

Comments:

The authors did not respond to comments the editor asked me to add to the original manuscript.

Best regards.

Author Response

Thanks to the reviewers for their suggestions. I have made amendments one by one according to your requirements.
